# Identification of 1600 replication origins in *S. cerevisiae*

**Eric J Foss, Carmina Lichauco, Tonibelle Gatbonton-Schwager, Sara J Gonske, Brandon Lofts, Uyen Lao, Antonio Bedalov***

Clinical Research Division, Fred Hutch Cancer Center, Seattle, United States

**Abstract** There are approximately 500 known origins of replication in the yeast genome, and the process by which DNA replication initiates at these locations is well understood. In particular, these sites are made competent to initiate replication by loading of the Mcm replicative helicase prior to the start of S phase; thus, 'a site that binds Mcm in G1' might be considered to provide an operational definition of a replication origin. By fusing a subunit of Mcm to micrococcal nuclease, we previously showed that known origins are typically bound by a single Mcm double hexamer, loaded adjacent to the ARS consensus sequence (ACS). Here, we extend this analysis from known origins to the entire genome, identifying candidate Mcm binding sites whose signal intensity varies over at least three orders of magnitude. Published data quantifying single-stranded DNA (ssDNA) during S phase revealed replication initiation among the most abundant 1600 of these sites, with replication activity decreasing with Mcm abundance and disappearing at the limit of detection of ssDNA. Three other hallmarks of replication origins were apparent among the most abundant 5500 sites. Specifically, these sites: (1) appeared in intergenic nucleosome-free regions flanked on one or both sides by well-positioned nucleosomes; (2) were flanked by ACSs; and (3) exhibited a pattern of GC skew characteristic of replication initiation. We conclude that, if sites at which Mcm double hexamers are loaded can function as replication origins, then DNA replication origins are at least threefold more abundant than previously assumed, and we suggest that replication may occasionally initiate in essentially every intergenic region. These results shed light on recent reports that as many as 15% of replication events initiate outside of known origins, and this broader distribution of replication origins suggest that S phase in yeast may be less distinct from that in humans than widely assumed.

*For correspondence:
abedalov@fredhutch.org

**Competing interest:** The authors declare that no competing interests exist.

## eLife assessment

This study represents a **valuable** addition to the understanding of the DNA replication origin selection process in the budding yeast. The authors provide **convincing** evidence that the number of possible origins of replication is much higher than previously appreciated, although many of the newly identified origins are likely to only direct replication initiation rarely. This work will be of interest to those studying DNA replication and investigating protein-DNA interactions across the genome.

## Introduction

DNA replication in *Saccharomyces cerevisiae* initiates at multiple sites across the genome known as replication origins. Fragments of DNA containing these sites were initially isolated in the late 1970s on the basis of their ability to support replication of plasmids (*Stinchcomb et al., 1979*; *Beach et al., 1980*; *Chan and Tye, 1980*; *Broach et al., 1983*), and in this context they acquired the designation of 'autonomously replicating sequences' or 'ARSs'. In subsequent years, additional origins were discovered using a variety of techniques to demonstrate their function (*Saffer and Miller, 1986*; *Brewer and*

*Fangman, 1987*; *Liachko et al., 2013*). Origins vary widely in their efficiency, with some being used in almost every cell cycle while others may be used in only 1 in 1000 S phases (*Boos and Ferreira, 2019*), with only the former being capable of supporting plasmid replication in the traditional ARS assay. The total number of reported yeast origins of replication is between 350 and 830 (*Wyrick et al., 2001*; *Xu et al., 2006Nieduszynski et al., 2007*; *Siow et al., 2012*).

The mechanism by which these sequences function as replication origins has been elucidated in great detail (*Fragkos et al., 2015*; *Bell and Labib, 2016*; *Costa and Diffley, 2022*; *Hu and Stillman, 2023*): During the G1 phase of the cell cycle, a six-protein 'origin recognition complex' (ORC) binds to the origin at a loosely defined 'ARS consensus sequence' (ACS) (*Bell and Stillman, 1992*) and loads a replicative helicase complex called MCM (for 'mini-chromosome maintenance'). MCM exists as two rings that encircle the DNA double helix, with each ring composed of six subunits encoded by MCM2-7 (*Bell and Botchan, 2013*; *Deegan and Diffley, 2016*). Mcm complexes are activated at the beginning of S phase through a combination of CDK and DDK kinase activities, and the two single hexamers then move in opposite directions to separate the two strands of DNA in front of the replication apparatus (*Francis et al., 2009*; *Deegan et al., 2016*; *Douglas et al., 2018*; *Lewis et al., 2022*). MCM is not loaded at every origin during every cell cycle, nor does every MCM complex that is loaded become activated (*Rhind and Gilbert, 2013*). This variation in both loading and firing leads to the variation in origin efficiency mentioned above.

Replication origins exhibit several characteristics that distinguish them from the surrounding DNA. First, they appear in nucleosome-free regions (NFRs) (*Lipford and Bell, 2001*; *Berbenetz et al., 2010*; *Eaton et al., 2010*). While nucleosomes are typically spaced at 165 base pairs, leaving approximately 18 base pairs of unbound DNA between, NFRs consist of longer (100–200 base pairs) stretches of DNA that are not only devoid or depleted of nucleosomes, but are also flanked on one or both sides by well-positioned nucleosomes (*Rando and Chang, 2009*; *Berbenetz et al., 2010*; *Eaton et al., 2010*; *Jansen and Verstrepen, 2011*). Second, these NFRs occur predominantly between rather than within genic regions (*Wang and Gao, 2019*). Finally, an evolutionary footprint marking the point of divergence of leading and lagging strand replication, which manifests itself as a skewing in the numbers of Gs versus Cs along a single strand of DNA, has been reported at replication origins. This is thought to reflect increased susceptibility of Cs to deamination when they are replicated in the more single-stranded context of the lagging strand (*Lobry, 1996*; *Grigoriev, 1998*). The association of patterns of GC skew with replication origins is more firmly established in prokaryotes than in eukaryotes, because the presence of a single replication origin and terminus makes this subtle signature easier to discern.

Until recently, limitations of population-based measurements of replication initiation precluded identification of very low activity origins. However, the development of single-molecule nanopore-based techniques such as Fork-seq and D-Nascent have overcome these constraints, and studies employing these techniques have revealed many initiation events between known origins (*Müller et al., 2019*; *Hennion et al., 2020*; *Theulot et al., 2022*). Indeed, the fraction of initiations that occur outside of known replication origins has been reported to be as high as 20% (*Hennion et al., 2020*). Given that the vast majority of these 'extra-origin' initiation events are detected as unique events in single-molecule studies, whether they reflect initiation from origins that are qualitatively similar to better characterized origins but are simply less active or, instead, reflect stochastic initiation events remains unclear. Thus, identification of further origins would benefit from a technique with higher throughput, to distinguish repeated use from stochastic events, while retaining the sensitivity required to distinguish very low activity origins from background noise.

We have previously shown that 'chromatin endogenous cleavage' ('ChEC') (*Schmid et al., 2004*; *Zentner et al., 2015*) allows one to identify sites at which Mcm double hexamers have been loaded with both high resolution, revealing that Mcm complexes are present immediately on or adjacent to the ACS, and also high sensitivity, with Mcm complexes identified at almost all known origins (*Foss et al., 2021*). The sensitivity of this technique is due, at least in part, to the fact that it creates relatively short fragments of interest within a background of much longer PCR-recalcitrant fragments: When Mcm subunits are tagged on their C-terminal ends with micrococcal nuclease (MNase), cleavage of the DNA underneath complexes loaded at origins will release small (50–100 base pairs) fragments of DNA in a background of much larger (tens of kb) inter-origin fragments. Because these small fragments are more amenable to PCR amplification steps, one can create sequencing libraries that are composed almost exclusively of the fragments of interest without size-based purification.

Here, we apply the ChEC technique to identify candidate MCM binding sites (CMBSs), and we then ask how deeply we can proceed into the lower intensity ChEC signals while still finding evidence that these sites serve as replication origins. Using replication itself as a readout, by making use of previously published data measuring the generation of single-stranded DNA (ssDNA) in S phase, we find evidence that the 1600 sites with the most intense ChEC signals are used as replication origins, with the magnitude of the ChEC signal corresponding to the level of ssDNA generated all the way down to the limits of detection of ssDNA. The next approximately 4000 peaks of ChEC signal continue to exhibit the hallmarks of replication origins described above, namely appearance in NFRs located in intergenic regions that are flanked by well-positioned nucleosomes and ACSs, and also a skewing of the numbers of Gs versus Cs. We conclude that low abundance sites of Mcm binding represent qualitatively similar, but less active, counterparts to better characterized origins of replication. This relaxation of the notion of what constitutes an origin of replication makes S phase in yeast resemble that in mammals more closely than has been previously thought.

## Results

### Identification of CMBSs that vary in abundance across three orders of magnitude

Our first goal was to determine, as comprehensively as possible, the genome-wide distribution of Mcm binding. To this end, we used MCM-ChEC, as previously described (*Foss et al., 2019*; *Foss et al., 2021*). Briefly, we fused the C-terminus of either MCM2, MCM4, or MCM6 to MNase, permeabilized cells, activated the MNase with calcium, and prepared sequencing libraries without size fractionation (*Figure 1A*). As previously reported, the size distribution of the DNA fragments produced by MCM-MNase cleavage peaked at 61 base pairs, which corresponds to the length of DNA covered by the MCM double hexamer in cryo-EM studies (*Figure 1B*). In order to capture the most robust peaks, we analyzed a total of 12 samples, collected under a variety of conditions, including G1 arrest, hydroxyurea (HU) arrest, and log phase. Measurements were highly reproducible, with mean and median values for $r^2$ of 0.98 and 0.99, respectively (*Figure 1—figure supplement 1*). Combining all 12 samples yielded a single data set with average coverage of 650×.

We next compared our results with previously published data derived from chromatin immunoprecipitation with anti-MCM antibodies, with or without treatment of the precipitated chromatin with exonucleases to trim the ends and thereby sharpen the signal (ChIP-seq and ChIP-exo-seq, respectively; *Figure 1C*; *Belsky et al., 2015*; *Das et al., 2015*; *Rossi et al., 2021*). Data sets derived from digestion with free MNase in G1-arrested cells are shown as controls (bottom rows within each panel) (*Foss et al., 2021*). In ChIP-exo, the signal on the Watson and Crick strands are shown separately, because each strand serves to delineate just one edge of the protein binding site: Specifically, the right and left sides of the binding site correspond to the steep declines on the right side of the Watson (blue) strand and the left side Crick (red) strand, respectively, and computational algorithms that recognize these patterns in the two strands are used to integrate the signals and thereby define binding site midpoints. *Figure 1C* shows the distribution of read depths on chromosome IV centered at ARS1 at four different scales. Both ChEC- and ChIP-based techniques identified a common set of sites (top left panel), which correspond largely to known origins. The higher sensitivity of ChEC over ChIP derives from the lower background signal, which is evident at all four scales. The advantage in resolution of ChEC over ChIP is most clear at the 3 kb scale: While both ChEC and ChIP reveal a footprint at the midpoint of Mcm binding (yellow shaded area in bottom right panel), the fraction of the total signal at ARS1 (red rectangle) that is derived from this midpoint is much higher in the former case. Our ability to determine the location of Mcm relative to the ACS and thereby to potentially infer the directionality of loading, or to detect evidence for DNA replication in the form of GC skew (described below), would not be possible without this high level of resolution. In summary, we conclude that ChEC reveals known sites of Mcm binding with a high degree of both sensitivity and resolution.

We next used a peak identification algorithm to convert our continuous genome-wide measurements of Mcm binding activity into a discrete list of CMBSs. Because a central goal of this study was to explore the behavior of low abundance Mcm binding sites, we wanted to avoid arbitrarily specifying an abundance cutoff to distinguish meaningful low abundance binding sites from background noise.

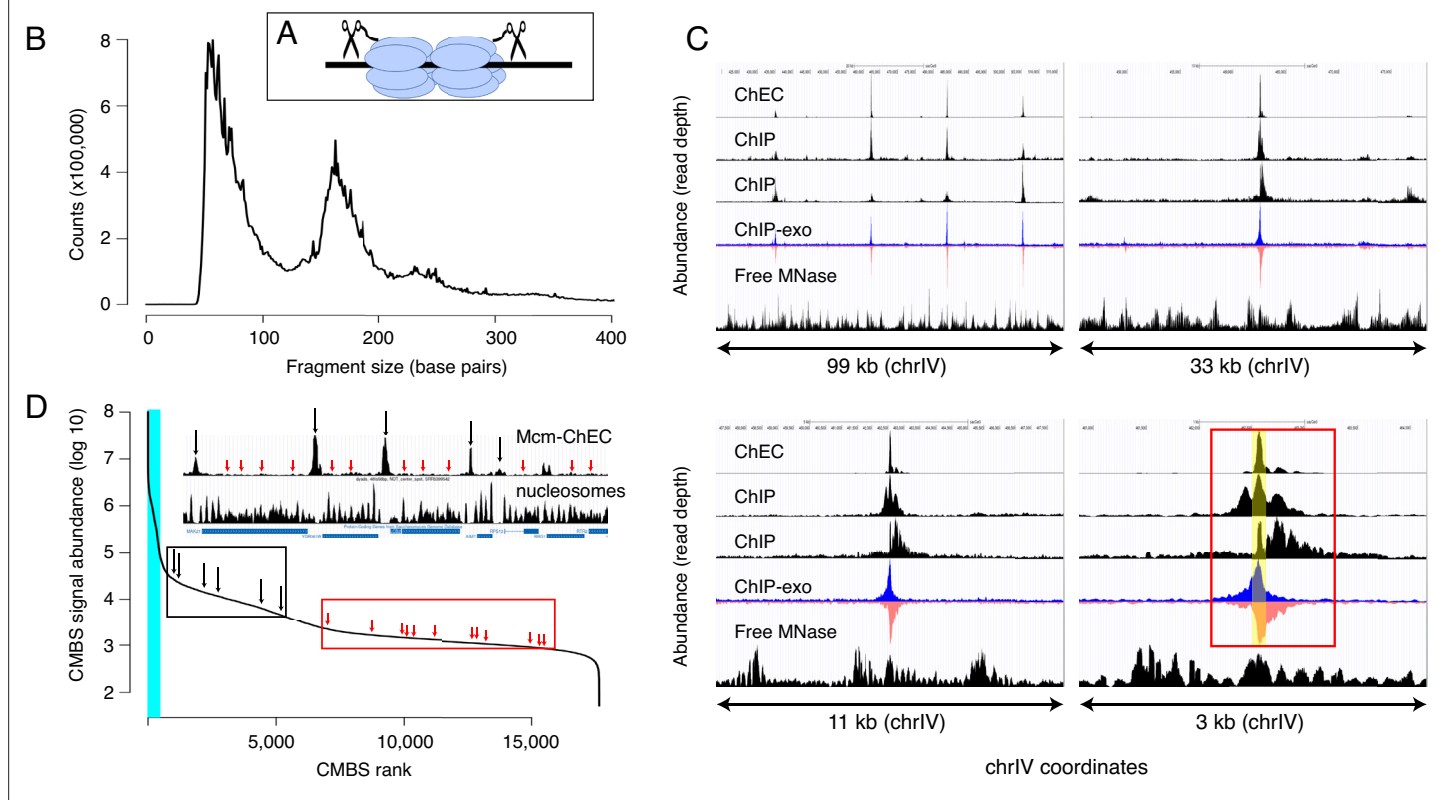

**Figure 1.** Mcm binding sites, identified by MCM-chromatin endogenous cleavage (ChEC), are consistent with those identified by Mcm-ChIP, and their occupancy varies over three orders of magnitude. (**A**) MCM-ChEC strategy relies on Mcm-microccocal nuclease (MNAse) fusion proteins that, when activated with exogenously added calcium, generate short DNA fragments corresponding to Mcm binding sites. These short fragments, which are conducive to PCR amplification, are identified through deep sequencing. (**B**) Library fragment size in MCM-ChEC library exhibits major and minor peaks at approximately 60 and 180 base pairs, corresponding to Mcm double helicases and nucleosomes, respectively. (**C**) Mcm binding sites identified by MCM-ChEC are consistent with results from Mcm-ChIP, but higher in resolution. Four panels show the same region of chrIV at various levels of magnification, from coordinates 417,855–516,737 (upper left) to coordinates 460,851–464,146 (lower right). The prominent peak in the middle corresponds to ARS416/ARS1. Yellow shading in lower right panel depicts within ARS1 (red rectangle) correspond to Mcm complex. From top to bottom, rows in each panel show MCM-ChEC (this study), Mcm- ChIP (**Belsky et al., 2015**), Mcm-ChIP (**Dukaj and Rhind, 2021**), Mcm-ChIP-exo (**Rossi et al., 2021**), and free MNase (**Foss et al., 2019**). (**D**) Signal intensities at candidate MCM binding sites (CMBSs) vary over approximately three orders of magnitude. The vertical portion of the plot at the far left represents ARS1200-1/ARS1200-2 in the repetitive rDNA. Blue vertical lines indicate the 142 peaks of MCM-ChEC signal that are within 100 base pairs of 1 of the 187 ARS consensus sequences (ACSs) reported in SGD. Inset shows an 11 kb region from chrIV, with MCM-ChEC signal in top row, nucleosome dyads in middle row, and gene locations in bottom row. Vertical black and red arrows point to CMBSs corresponding to intergenic Mcm helicases and low level background signal from nucleosomes, respectively, with the ranks of these sites depicted by corresponding arrows on the main curve.

The online version of this article includes the following source data and figure supplement(s) for figure 1:

**Source data 1.** Data used to generate *Figure 1B*.

**Source data 2.** Data used to generate *Figure 1C*.

**Source data 3.** Data used to generate *Figure 1D*.

**Figure supplement 1.** MCM-chromatin endogenous cleavage (ChEC) measurements are reproducible.

**Figure supplement 1—source data 1.** Data used to generate *Figure 1—figure supplement 1*.

Instead, we chose to compile an exhaustive list of CMBSs, assigning approximately one per kb for the entire genome, and then retrospectively determine how far down this list we could go while still discerning known characteristics replication origins.

We used a simple iterative algorithm to generate a list of CMBSs, first creating a genome-wide per-base pair read depth matrix based on our 12 MCM-ChEC data sets and smoothing these values with a 200 base pair moving window. The highest peak in this data set was assigned a rank of 1. To avoid assigning multiple IDs to essentially the same peak, once a CMBS was assigned, we

eliminated the region within 500 base pairs of the midpoint of that peak from consideration for the definition of further peaks before repeating the process to assign a CMBS with a rank of 2, etc., ultimately arriving at a list of 17,618 candidate sites (for details, see Materials and methods). CMBS signal magnitude exhibited a rapid decline over the course of the first 500 CMBSs, followed by a relatively gradual decline, punctuated by a slight inflection around 5500 (*Figure 1D*). As discussed below, this inflection point coincides approximately with the point at which CMBSs no longer exhibit characteristics of replication origins. There were high abundance CMBSs (ranks ≤ 500) within 100 base pairs of 82% of origins reported in SGD (282 out of 343 unambiguously mapping origins). We note, however, that the regions defined as origins of replication in SGD tend to be significantly larger (mean size 290 base pairs) than the 61 base pairs occupied by an Mcm double hexamer. More precise origins' midpoint coordinates can be assigned for that subset of origins that have a defined ACS (187 of 343), and in these cases, there were CMBSs within 100 base pairs of 78% (146 out of 187) of these ACSs. These 146 CMBSs, marked by blue vertical line in *Figure 1D*, were all ranked among the most abundant 500. Thus, whether we use relaxed or stringent criteria for the coordinates of reported origins, we conclude that the vast majority of known origins are associated with the 500 most abundant CMBSs.

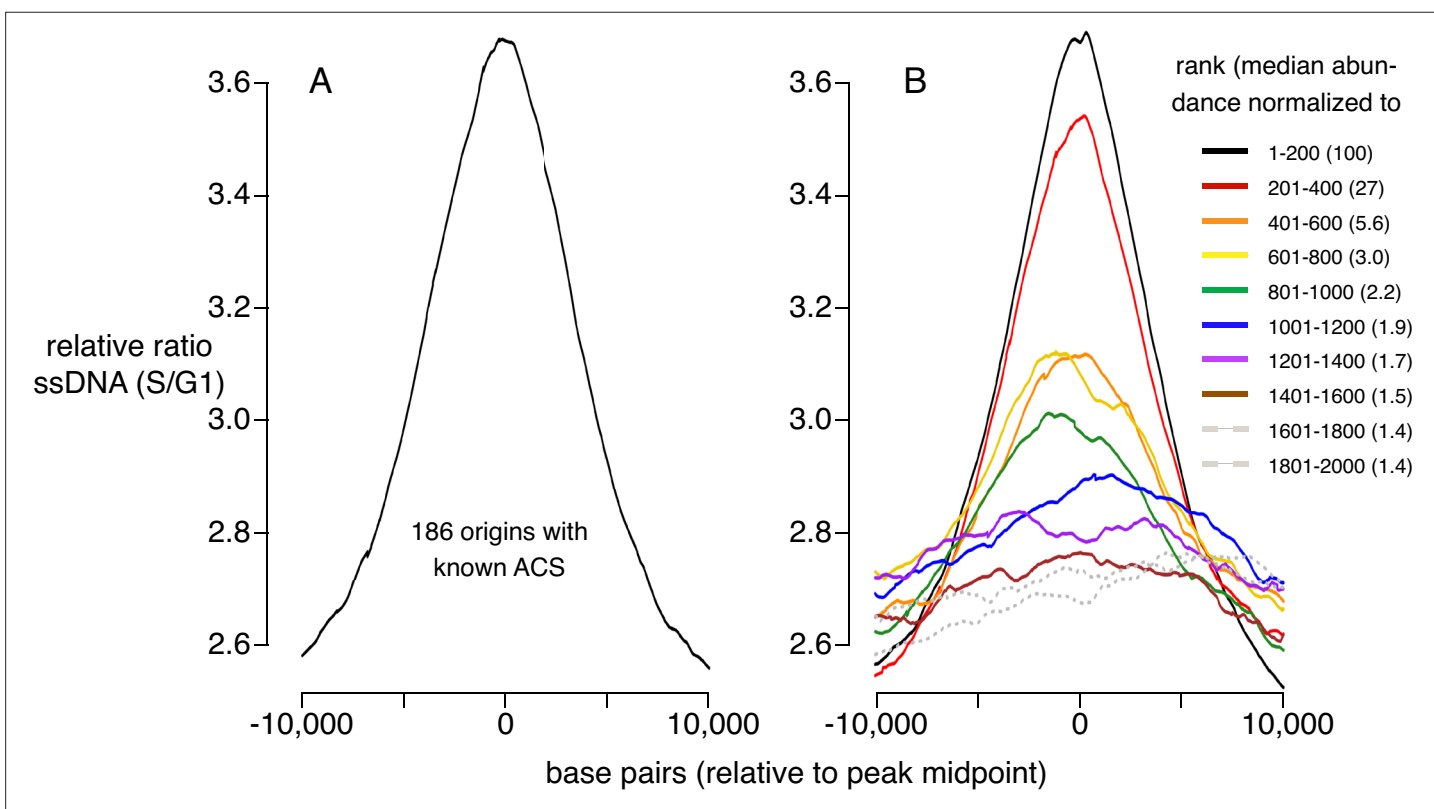

**Figure 2.** Replication activity at known origins and Mcm binding sites. Replication activity was assessed by generation of single-stranded DNA (ssDNA) in *rad53* mutants released from G1 arrest into medium containing hydroxyurea (HU). Plots show relative ratio of ssDNA in S/G1, as described (*Feng et al., 2006*). (**A**) Average replication profile centered on 181 ARS consensus sequences (ACSs) (all ACSs at least 10 kb from chromosome end listed in SGD for which data were available in *Feng et al., 2006*). (**B**) Average replication profiles of groups of 200 of the 2000 most prominent peaks of MCM-chromatin endogenous cleavage (ChEC) signal, centered on peak midpoints. Numbers in parentheses indicate median abundance of Mcm-ChEC signal, normalized such that the median signal in the highest group is 100.

The online version of this article includes the following source data and figure supplement(s) for figure 2:

**Source data 1.** Data used to generate *Figure 2B*.

**Figure supplement 1.** Replication activity and licensing at Mcm binding sites, assessed by single-stranded DNA (ssDNA) generation, BrdU incorporation, and qPCR.

**Figure supplement 1—source data 1.** Data used to generate *Figure 2—figure supplement 1*.

## Assessment of replication initiation at CMBSs

To determine how CMBSs correspond to sites where DNA replication initiates, irrespective of what has previously been defined as a replication origin, we used published data measuring the appearance of ssDNA during S phase in cells released from an alpha factor G1 block into medium containing HU, which impedes replication fork progression by inhibiting ribonucleotide reductase (*Feng et al., 2006*). To establish a benchmark for replication initiation, we first analyzed ssDNA accumulation at 181 well-characterized origins centered on their known ACSs and, as expected, we observed a peak of ssDNA accumulation (*Figure 2A*). Furthermore, this peak was similar in both shape and magnitude to an analogous peak generated by juxtaposing the midpoints of the 200 most abundant CMBSs (*Figure 2B*, black line), which is expected, because the two groups are largely overlapping. Extending this line of analysis, we asked how far down in abundance we could go, in groups of 200, and still discern a signal of replication initiation (*Figure 2B*, colored lines). Two aspects of these results are notable: First, peaks of ssDNA signal, as judged by higher signal at the midpoints than the edges, extend down to the eighth cohort (brown line), which corresponds to CMBSs ranked 1401–1600. Of course, this does not imply that all of these sites function as replication origins, nor does it imply that no sites below that rank do so, since we have reached the limits of detection of this ssDNA-based assay. Nonetheless, it suggests that replication activity is common among sites extending at least down to rank 1600. Second, there is a rapid drop in ssDNA after the second cohort, demonstrating that the sharp drop in Mcm abundance shown in *Figure 1D* is recapitulated in a drop in replication activity. We have previously used southern blotting to demonstrate that approximately 90% of the DNA at one of the most active known origins (ARS1103) is cut by Mcm-MNase (*Foss et al., 2021*), and to thereby infer that 90% of cells have a double helicase loaded at this origin. Combining this measurement with six additional measurements of licensing in cohort 1, we used linear regression ($r^2$=0.7) to infer a median value of 69% for cohort 1 (*Figure 2—figure supplement 1*). Because the median abundance in the eighth cohort is 1.5% of that in the first cohort, we estimate that CMBSs in the eighth cohort are typically licensed in 1% of cells in the population (69% × 0.015=1.0%). *rad53* mutants are used here because they do not suppress firing of late origins, and therefore allow a more comprehensive assessment of origin activity. ssDNA measurements for both wild type and *rad53* genotypes at other time points (*Figure 2—figure supplement 1*), as well as BrdU-based measurements of replication activity in *rad53* mutants (*Yoshida et al., 2014*; *Figure 2—figure supplement 1*), yielded comparable results. We conclude that replication activity parallels Mcm abundance, and that this activity continues to be prevalent at least through the top 1600 CMBSs.

## Mcm binding sites are enriched in NFRs

The abundance of CMBSs that rank below the limit of detection of ssDNA is only marginally lower than that in the cohort that still exhibits ssDNA accumulation (e.g. 1.5% vs 1.4% for cohorts 1401–1600 vs 1601–1800). To determine whether these lower abundance sites, whose potential origin activity was below the limits of detection by ssDNA, might, nonetheless, function as replication origins, we assessed them for another hallmark of replication origins, namely their presence in intergenic NFRs. Nucleosome positions were determined based on published data that exploit site-directed mutagenesis to confer chemical cleavage activity to a histone subunit (*Chereji et al., 2018*). Specifically, residue 85 of histone H3 was changed from a glutamine to a cysteine. With two copies of H3 per histone octamer, each nucleosome ends up containing two of these engineered cysteines, with the two cysteines separated by a 51 base pair stretch of the DNA backbone centered on the nucleosome dyad. Ex vivo coupling of these residues to phenanthroline in the presence of copper ions and peroxide endows these cysteines with endonuclease activity, thereby releasing a 51 base pair fragment of DNA centered at the midpoint of the nucleosome, where the DNA is most tightly wrapped. High-throughput sequencing, followed by read depth profiling of library fragments in the 51 base pair range (46–56 base pairs), thus provides a high-resolution map of nucleosome positioning that is relatively robust to variable cleavage of the DNA due to unwrapping at nucleosome entry and exit sites. Heat maps of these data centered at the midpoints of the CMBSs reveal an approximately 150 base pair NFR, consistent with the size of NFRs at the known origins (*Berbenetz et al., 2010*; *Eaton et al., 2010*), for the first 5000 CMBSs and their flanking nucleosomes (*Figure 3A*). In contrast, we observed a clear nucleosomal signal at the midpoints of most CMBSs that rank >5000. To better characterize this transition from NFR to nucleosome, we analyzed cumulative chemical cleavage signal in cohorts

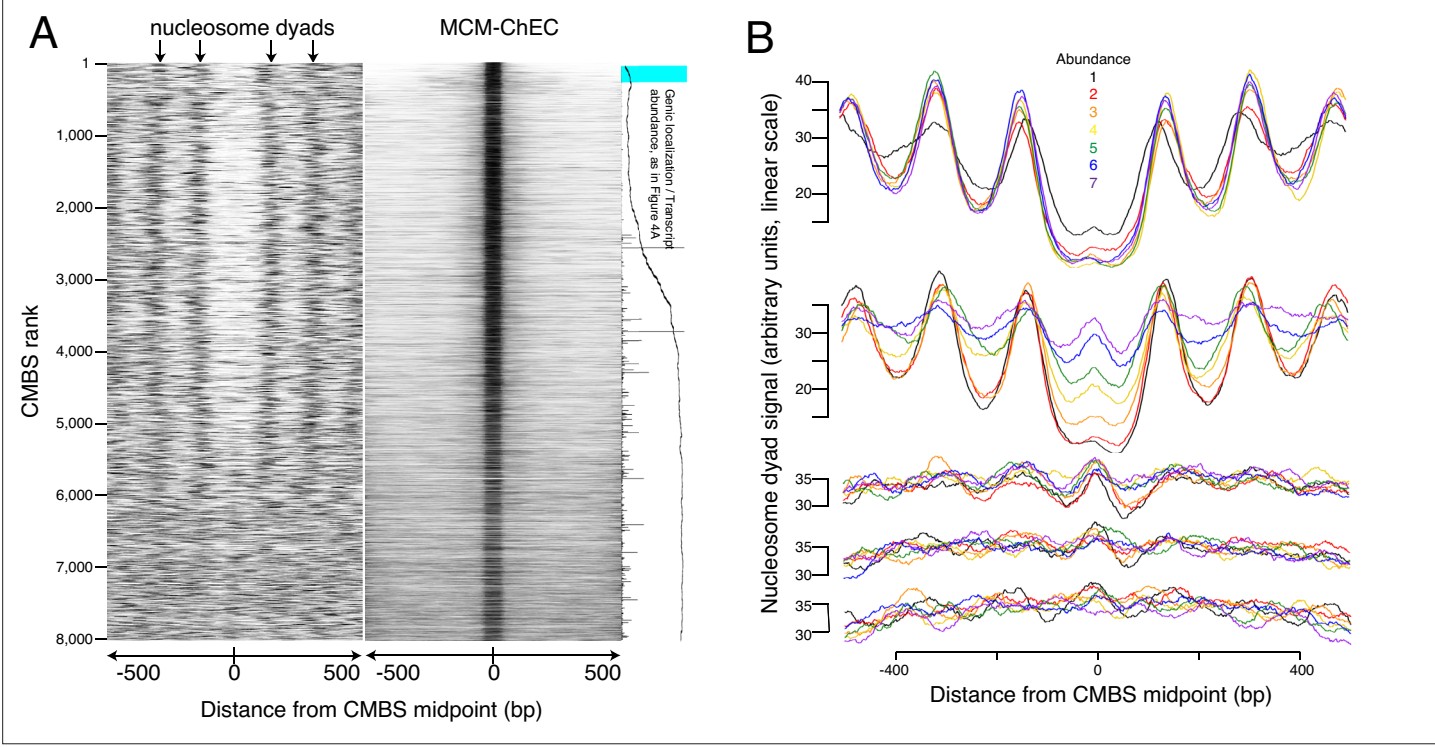

**Figure 3.** Most abundant 5500 peaks of MCM-chromatin endogenous cleavage (ChEC) signal are located predominantly in nucleosome-free regions. (**A**) Left heat map shows 46–56 base pair fragments marking nucleosome dyads that were generated by ex vivo nuclease activity of H3Q85C coupled to phenanthroline (*Chereji et al., 2018*). Right heat map shows 51–100 base pair library fragments from MCM-ChEC. Both heat maps are centered on midpoints of the top 8000 MCM-ChEC peaks, ranked by abundance. Each row corresponds to a single peak, with the most abundant peak at the top. A portion of *Figure 4A* is shown rotated by 90° (right) to indicate average locations of 50 candidate MCM binding sites (CMBSs) relative to gene bodies (main curve), transcription levels of genes in those cases when CMBSs are genic rather than intergenic (horizontal bars), and CMBSs within 100 base pairs of an ARS consensus sequence (ACS) listed in SGD (light blue). (**B**) Composite signals of nucleosome dyads for groups of 500 CMBSs. The x axis is identical to that in the heat maps in (**A**). Each row shows 7 groups of 500. Moving from the top row to the bottom row corresponds to moving from high abundance CMBSs to low abundance CMBSs. Within each row, the order from high abundance to low abundance is indicated by color, as noted in the figure. The y axes are linear scales showing the relative abundance of nucleosomal signal, and the scales of these axes are the same for each row.

The online version of this article includes the following source data and figure supplement(s) for figure 3:

**Source data 1.** Data used to generate *Figure 3A* and *Figure 3—figure supplement 1*.

**Source data 2.** Data used to generate *Figure 3B*.

**Figure supplement 1.** Figure identical to *Figure 3*, except that nucleosomes were identified by micrococcal nuclease (MNase) treatment of chromatin.

of 500 successively ranking CMBSs centered at their midpoint (*Figure 3B*). This analysis revealed a central valley of ChEC signal corresponding to the NFR in the first 10 cohorts (through 4501–5000), at which point we observe a gradual transition from central NFR to a predominantly nucleosomal signal at the midpoint, which is complete by the 14th cohort (ranks 6501–7000). These findings indicate that most of the low abundance (rank >5000) CMBSs correspond to nucleosomes, presumably due to low-level non-specific MNase activity. Mapping of nucleosome locations by enzymatic digestion of chromatin with MNase (*Foss et al., 2021*) confirmed these results (*Figure 3—figure supplement 1*). It is notable that, when Mcm is present, it is present predominantly as a single double hexamer (based on the size of the protected region in the right panel of *Figure 3A*), and that this remains true across the entire range of abundance shown in *Figure 3A*. This argues against models in which higher replication activity at more active origins is caused by the loading of more Mcm double hexamers at those origins within a single cell, since such models predict that multiple Mcm footprints should be more prevalent at the top (high abundance) of the Mcm-ChEC heat map in *Figure 3A* than at the bottom. We conclude that nucleosome mapping studies using two orthogonal methods demonstrate a fundamental difference between the first 5000 CMBSs, which are located in the NFR and are flanked by adjacent nucleosomes, from those that rank >5000, which largely correspond to nucleosomes.

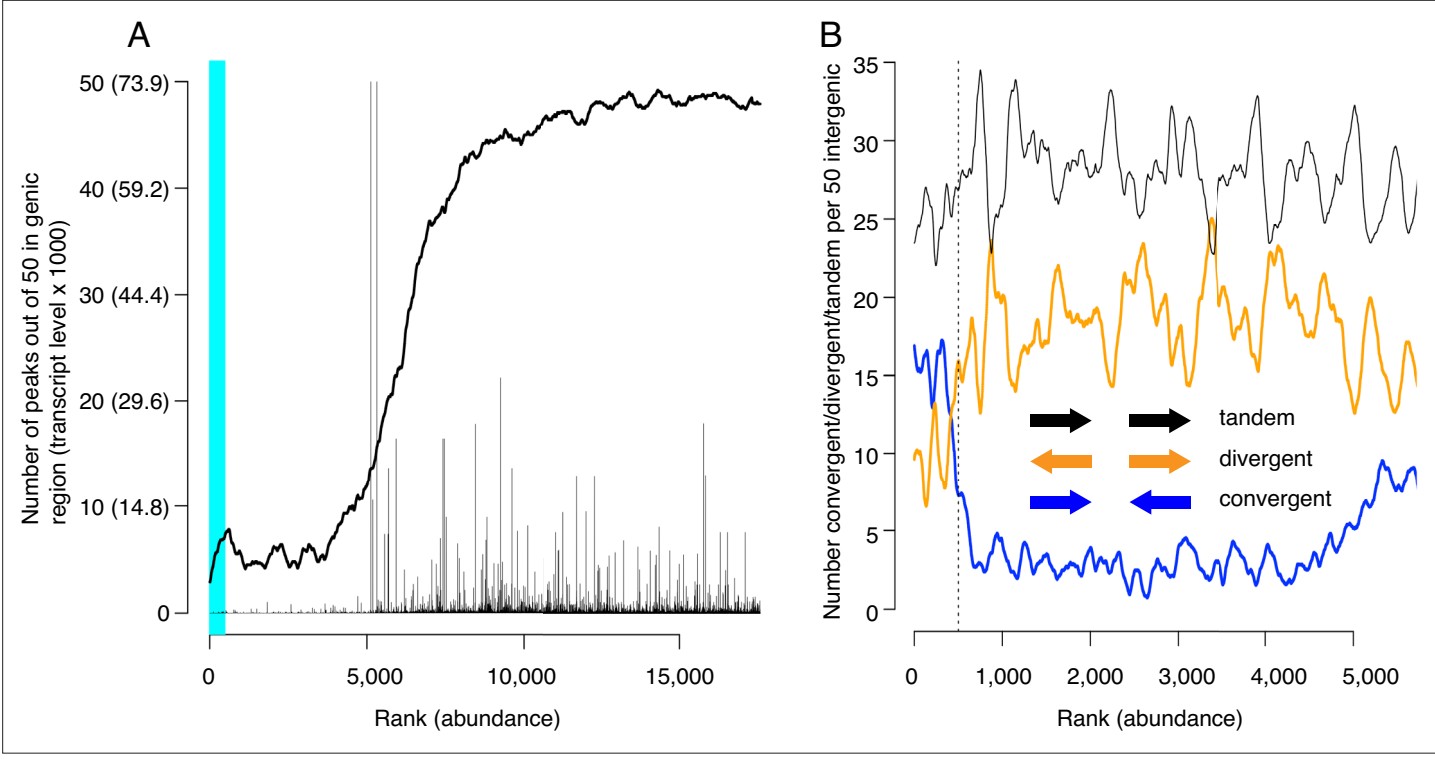

**Figure 4.** Locations of candidate MCM binding sites (CMBSs) with respect to locations and both levels and directions of transcription of adjacent genes. (**A**) Continuous curve shows number of CMBSs, from a sliding window of 50, that are located within gene bodies. Vertical lines show levels of transcription of genes in those instances in which the corresponding CMBS is genic. Light blue area indicates CMBSs within 100 base pairs of ARS consensus sequence (ACS) listed in SGD. X axis is arranged by CMBS abundance rank, from high abundance (left) to low abundance (right). (**B**) Orientation of transcription of flanking genes in those instances when CMBSs are intergenic. Plots show number of convergently (blue), divergently (orange), and tandemly (black) transcribed genes enumerating only those CMBSs, within a sliding window of 50, that are intergenic. X axis is limited to CMBSs with ranks ≤ 5500, because higher ranks are mostly genic, and therefore cannot be classified according to relative transcription directions of flanking genes.

The online version of this article includes the following source data for figure 4:

**Source data 1.** Data used to generate *Figure 4A*.

**Source data 2.** Data used to generate *Figure 4B*.

## Mcm binding sites localize to intergenic regions or within poorly transcribed genes

We next evaluated CMBSs for another feature of replication origins, namely their localization to intergenic regions. As expected, the most abundant sites, which contain most of the known origins of replication, were located predominantly between rather than within genes (89% of sites ranked ≤ 500 were intergenic). Furthermore, in those instances when high abundance sites were located within genes, those genes were frequently non-transcribed. For example, 15 of the 56 genes that contained a high abundance site have been implicated in meiosis and sporulation and are not expressed during vegetative growth (~5 out of 56 expected from random sampling), consistent with previous observations (*Mori and Shirahige, 2007*; *Blitzblau et al., 2012*). CMBSs remained largely intergenic with decreasing abundance until approximately rank 5000, at which point they switched rather abruptly to genic localization (*Figure 4A*). Moreover, the level of transcription of genes that contained CMBSs showed an abrupt increase at this same point. This coincides approximately with the point at which sites transition from nucleosome-free to nucleosome-occupied regions. Thus, the first 5000 ranking CMBSs, in addition to being located in the NFR, are also largely intergenic, or located within poorly transcribed genes, whereas the CMBSs that rank >5000 correspond largely to nucleosomal signal within transcribed genes. Such a distribution of ChEC signal can be appreciated throughout the genome, with most intergenic fragments featuring a clear high-ranking ChEC signal within their NFR

(black arrows in *Figure 1D*), whereas the low abundance signal within gene corresponds to nucleosomes (red arrows in *Figure 1D*).

Inspection of the orientation of genes that flank CMBSs supports the simple view that Mcm complexes are loaded at these sites rather than, for example, being loaded within the body of adjacent genes and then pushed out into the intergenic region by RNA polymerase. Lower abundance intergenic MCM signals (ranks between 500 and 4000) showed a distinct preference for divergent over convergent transcription of flanking genes (*Figure 4B*). This is the opposite of what would be expected if these helicase complexes were loaded within gene bodies and then pushed out by transcription. Furthermore, this is consistent with our observation that these sites are flanked by ACSs (see below). In contrast to the low abundance sites, the most abundant 500 sites showed a preference for convergent over divergent transcription (left of vertical dotted line in *Figure 4B*), in agreement with a previous report (*MacAlpine and Bell, 2005*; *Li et al., 2014*). While this might be interpreted to suggest displacement of helicase complexes by transcription, our published ChEC analysis of Mcm locations at origins with defined ACSs, all of which fall within the most abundant 500 sites, demonstrates that these helicases are found almost exclusively immediately adjacent to the ACS. In summary, we favor the parsimonious view that Mcm complexes are initially loaded within the same NFRs in which we have localized them through ChEC.

## Mcm binding sites are flanked by ACSs, and Mcm is loaded downstream of known ACSs

Another characteristic of known origins in *S. cerevisiae* (although not in most other organisms) that we could use as a criterion to assess the nature of Mcm binding sites is the presence of an ACS (*Broach et al., 1983*; *Kearsey, 1984*; *Srienc et al., 1985*; *Deshpande and Newlon, 1992*). The ACS is a loosely defined AT-rich 11–17 base pair sequence whose directionality is determined on the basis of the relative numbers of As versus Ts in the two strands (*Breier et al., 2004*). This sequence is recognized by the Orc, a six-protein complex that loads MCM (*Bell and Stillman, 1992*; *Newlon and Theis, 1993*; *Berbenetz et al., 2010*; *Eaton et al., 2010*; *Singh and Krishnamachari, 2016*). Although the characteristics that allow a sequence to serve as an Orc binding site remain poorly understood, with, for example, only approximately half of the origins listed in SGD reporting associated ACS signals, researchers have calculated a probability weight matrix to evaluate the fidelity of the correspondence of any 17 base pair sequence with a hypothetically 'perfect' ACS (*Coster and Diffley, 2017*). Because the ACS is asymmetric, these scores are calculated separately for the Watson and Crick strands (shown in blue and red, respectively, in *Figure 5A*). Using this metric, we found that the most abundant 5500 CMBSs are enriched for flanking ACSs, approximately 35 base pairs from the midpoint of Mcm binding. Furthermore, these ACSs are oriented in a direction consistent with that expected from in vitro loading of Mcm complexes by Orc. Finally, the degree to which these sequences match a hypothetically perfect ACS decreases with increasing rank (i.e. with decreasing abundance), largely disappearing by rank 6000 (*Figure 5B*). Our results allowed us to explore the question of whether individual Mcm binding sites contain two Orc binding sites or whether, instead, the cumulative high scores on both sides of CMBSs in *Figure 5A* reflect the juxtaposition of multiple unoriented sites, each of which contains just one Orc binding site. This issue has implications for the mechanism by which Orc loads two hexameric helicases and for the design of biochemical experiments aimed at elucidating that mechanism (*Coster and Diffley, 2017*; *Gupta et al., 2021*). To address the question, we extracted 594 CMBSs that had a particularly strong (PWM ≥ 10.8) ACS on the Watson strand and then asked whether there was still an enrichment for a corresponding ACS on the Crick strand, as shown in *Figure 5C*. We found that 594 loci with a strong ACS on the Watson strand were still enriched for ACS on the Crick strand, thus supporting models in which origins have two binding sites for Orc (*Coster and Diffley, 2017*).

The high resolution of our data also allowed a more demanding test of the directionality of loading of MCM relative to the ACS: The results described above demonstrate that sequences centered on MCM binding sites will, in aggregate, contain adjacent ACSs that are oriented in the direction predicted by biochemical experiments. Conversely, and more stringently, we could ask whether, when centering and orienting sequences according to previously reported ACSs, the peak of MCM signal appeared on the expected side of the ACS. Consistent with directional MCM loading downstream of the ACS, cumulative abundance of Mcm-ChEC signal at 187 origins centered at their ACS peaked

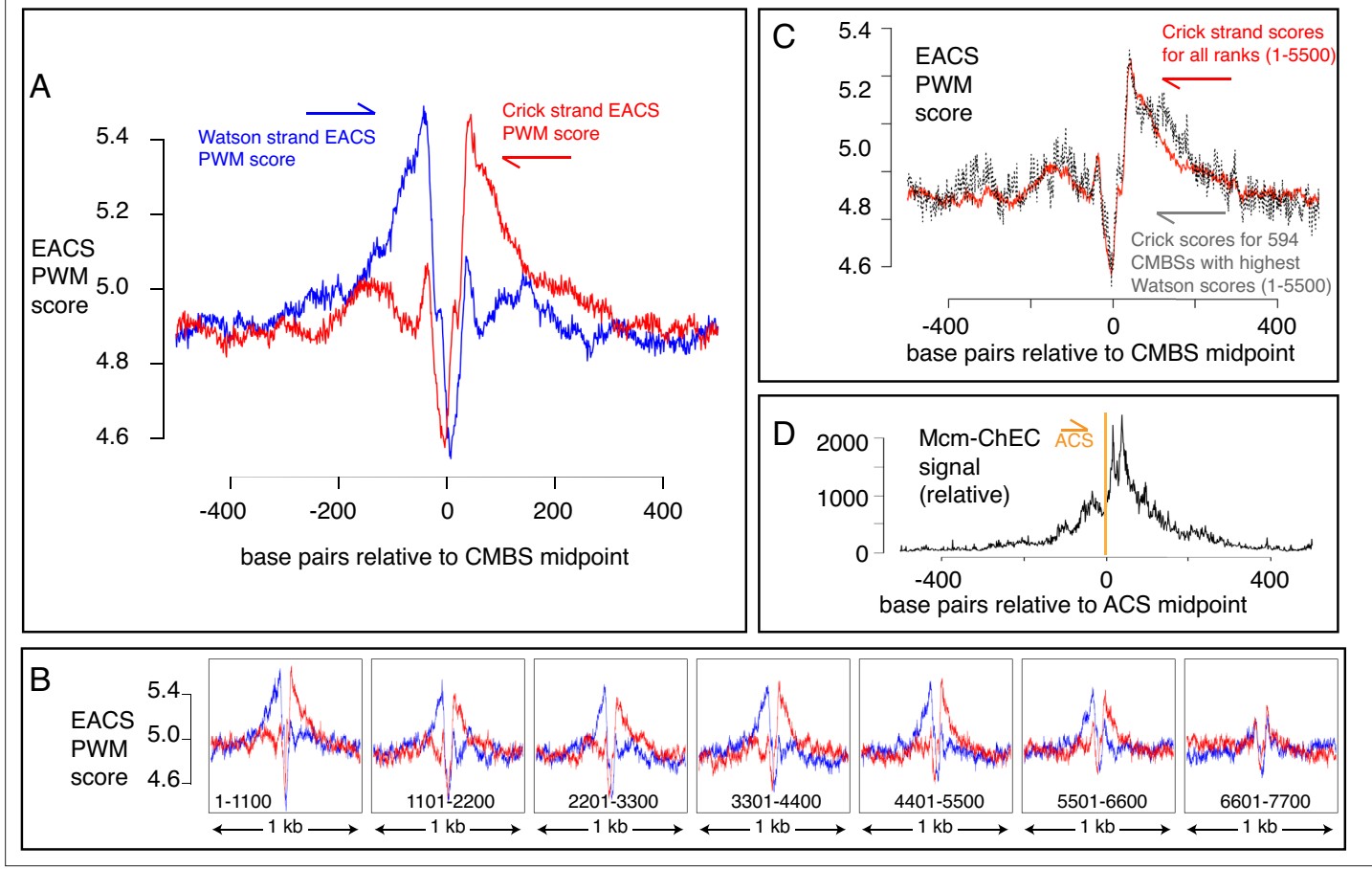

**Figure 5.** EACS PWM scores indicate that peaks of MCM signal are flanked by ARS consensus sequences (ACSs), and that Mcm helicases are loaded downstream of the ACS (maximum possible score is 12.742). (**A**) Composite scores for Watson (red) and Crick (blue) strands for candidate MCM binding sites (CMBSs) with ranks ≤ 5500. (**B**) Composite scores for Watson (red) and Crick (blue) strands for CMBSs with ranks ≤ 7700, shown in groups of 1100. (**C**) Average EACS-PWM scores on Crick strand for peaks with the top 10% of EACS-PWM scores on the Watson strand are as high as average scores on the Crick strand for all 5500 peaks of MCM-chromatin endogenous cleavage (ChEC) signal. 10% of peaks with highest EACS-PWM scores on Wason strand are shown in black and total peaks are shown in red. This result supports a model in which peaks of Mcm signal are flanked on both sides by ACS signals, because, even if peaks have a very high EACS-PWM on the left side, they still maintain a significant score on the right side. (**D**) Cumulative MCM-ChEC signal for 187 ACSs indicate that Mcm helicases are loaded downstream of the ACS. 1 kb windows of MCM-ChEC signals centered on each of the 187 ACSs listed in SGD were oriented according to the ACS, normalized to 100, and then added to create a cumulative sum vector showing the location of MCM-ChEC signal relative to the ACS. MCM-ChEC signal was quantified according to the midpoints of library fragments with inserts from 51 to 100 base pairs.

The online version of this article includes the following source data for figure 5:

**Source data 1.** Data used to generate *Figure 5A, D* .

**Source data 2.** Data used to generate *Figure 5B*.

**Source data 3.** Data used to generate *Figure 5C*.

downstream of the ACS (*Figure 5D*). There were CMBSs within 100 base pairs of 146 of the 187 ACSs reported in SGD, and in 112 out of 146 cases (77%), the CMBS was downstream of the ACS. This in vivo confirmation of the in vitro prediction for directionality has not been previously possible (*Belsky et al., 2015*; *Dukaj and Rhind, 2021*), as it requires a level of resolution not attainable by ChIP-based techniques (*Figure 1B*). In interpreting the results above, it is important to remember that the locations at which we are detecting Mcm complexes by ChEC do not necessarily reflect the locations at which those complexes were loaded, since Mcm double hexamers can slide along the DNA after loading (*Rando and Chang, 2009Gros et al., 2015*; *Foss et al., 2019*). Furthermore, in the case of ARS1, two reports have demonstrated a requirement for the B2 element for Mcm loading, though not for Orc binding, suggesting that Orc may bind to the ACS but then load Mcm at the B2 element

(*Zou and Stillman, 2000*; *Lipford and Bell, 2001*). This would still leave Mcm loaded downstream of the ACS, but we note this result to emphasize that not all details of Mcm loading in vitro have been definitively established. In summary, our results not only provide in vivo support for in vitro predictions of the directionality of Mcm loading by Orc, but they are also consistent with the notion that essentially all of the most abundant 5500 Mcm binding sites can function, at least on rare occasions, as replication origins.

## GC skew as evolutionary footprint of replication initiation

GC skew, which is a measure of the deviation from the expected 1:1 ratio of Gs to Cs along a single strand of DNA, suggests another potential metric to determine whether sites have been used as replication origins. GC skew is calculated as (G-C)/(G+C), with G and C representing the numbers of Gs and Cs, respectively, within a specified window sliding along a single strand of DNA (*Lobry, 1996*; *Grigoriev, 1998*). Consistent use of particular sequences as replication origins over long time scales can leave evolutionary 'footprints' in which an excess of Gs over Cs reverses to become an excess of Cs over Gs as one traverses the site of initiation of replication. Such reversals of GC skew reflect the combination of: (1) the transition between leading and lagging strand synthesis precisely

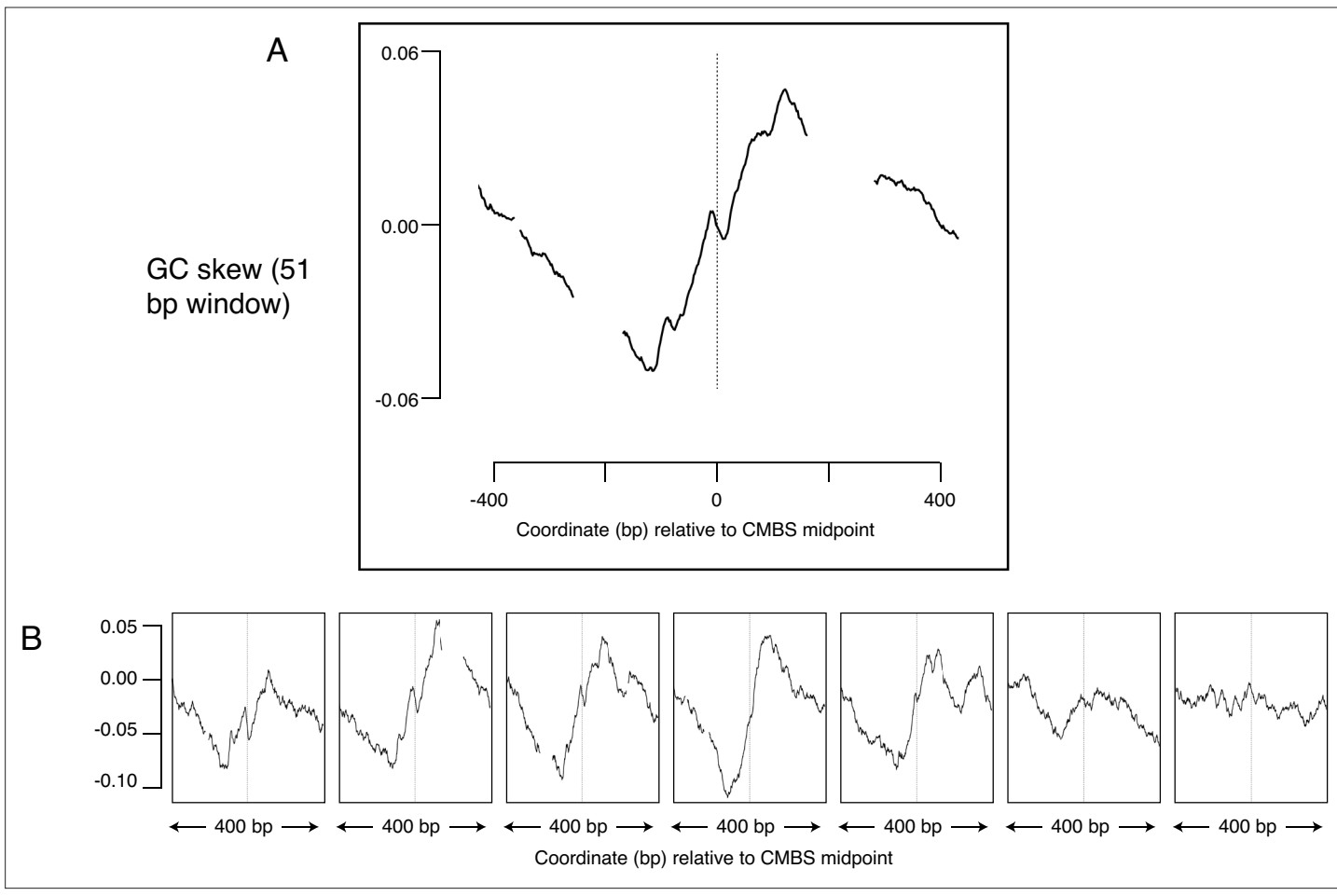

**Figure 6.** Average GC skew increases across the midpoints of the most abundant 5500 peaks of MCM-chromatin endogenous cleavage (ChEC) signal. GC skew was calculated according to the numbers of Gs and Cs in the Watson strand along a 51 base pair moving window, as (G–C)/(G+C); the skew value plotted corresponds to the score at the midpoint of that window. (**A**) Average EACS PWM scores for the most abundant 5500 peaks of MCM-ChEC signal. (**B**) Averages of GC skew for groups of 1100, arranged from high (left) to low (right) abundance.

The online version of this article includes the following source data and figure supplement(s) for figure 6:

**Source data 1.** Data used to generate *Figure 6* and *Figure 6—figure supplement 1*.

**Figure supplement 1.** Comparison of pattern of GC skew when centered on ARS consensus sequences (ACSs) versus candidate MCM binding sites (CMBSs).

where DNA replication initiates; (2) the fact that the template for lagging strand synthesis exists in a single-stranded state more than does the template for leading strand synthesis; (3) the increased vulnerability of cytosines to deaminases when in a single-stranded state; and (4) replication-induced conversion of deaminated cytosine to thymine. This phenomenon has been best characterized in bacteria, where the presence of a single origin and terminus of replication on each circular chromosome ensures that, at any particular location in the genome, each strand of DNA consistently serves as a template for exclusively either the leading or the lagging strand polymerase (*Lu and Salzberg, 2020*). A change in GC skew across origins has also been reported in yeast (*Li et al., 2014*), but the magnitude of the effect is small, and the direction of the effect is contrary to both the observation in bacteria and the theoretical expectation (*Figure 6—figure supplement 1*). To determine whether sites identified by MCM-ChEC exhibited patterns of GC skew consistent with their use as replication origins, we aligned 400 base pair sequences centered on the 5500 most abundant CMBSs and calculated GC skew using a 51 base pair sliding window. This analysis revealed a general rise in GC skew as one follows either strand in the 5' to 3' direction across the CMBS, a result that is consistent with both the theoretical expectation and the empirical observation in bacteria (*Figure 6A*). Subdividing CMBSs into groups of 1100 and arranging these groups according to decreasing abundance revealed that GC-skew-based evidence for DNA replication initiation continued until approximately rank 6000 (*Figure 6B*). Our ability to detect this subtle effect requires precise alignment at the midpoint of Mcm double-hexamer binding, as alignment of identical sequences according to the ACS does not recapitulate the result (*Figure 6—figure supplement 1*). A short inversion of this pattern at the midpoint is consistent with biochemical results showing that the two Mcm hexamers cross over each other before they begin diverging (*Douglas et al., 2018*) (see Discussion). We conclude that the distortion of the expected 1:1 ratio of Gs to Cs on a single strand of DNA is consistent with the hypotheses that the most abundant 6000 CMBSs have served as replication origins.

## Discussion

The first yeast origin of replication, ARS1, was reported in 1979 as a sequence that could sustain plasmid replication (*Stinchcomb et al., 1979*). Not surprisingly, given the assay by which it was discovered, ARS1 is among the most active origins in the genome. Over the subsequent 25 years, this simple assay for origin activity led to numerous other genomic sequences being designated as origins, with the number stabilizing at around 500 (*Nieduszynski et al., 2007*; *Siow et al., 2012*). While few of these origins are as efficient as ARS1, all must exhibit replication activity sufficient to support colony formation. For example, origins that fire in only 1 in 10 S phases will not produce even a microcolony in a plasmid transformation assay, and therefore will not be detected. Thus, the set of known origins is largely restricted to those with relatively high activity. This conclusion is consistent with *Figure 1D*, in which the known (light blue) origins are strikingly concentrated on the left (high abundance) side of the curve.

We used MCM-ChEC to identify a more comprehensive set of replication origins. In this approach, we first identify sites at which Mcm helicases are loaded and then subsequently examine those sites for characteristics of replication origins. These characteristics include, but are not limited to, replication itself. There are four advantages to this approach: First, the ChEC assay is exceptionally sensitive because, as noted in the Introduction, the short DNA fragments of interest enjoy a tremendous selective advantage for amplification during PCR steps involved in library construction. Thus, we can detect Mcm complexes that are loaded in as few as 1 in 500 cells (*Foss et al., 2021*), even though such low affinity Mcm binding sites are not expected to be capable of supporting autonomous replication of a plasmid. Second, by averaging signals of replication from multiple Mcm binding sites, we were able to extract weak signals of replication. This is due to the fact that noise, which is randomly distributed, will tend to cancel itself out, while signals of replication will consistently augment the signal at the midpoint of the origin (*Figure 2*). An inevitable shortcoming to this approach is that it precludes analysis of specific sites; in other words, not every member of the group will share the average characteristic of that group. Third, even beyond the level of detection possible through direct measurements of replication, such as generation of ssDNA, we were able to infer past replication activity from the subtle evolutionary signatures of replication that manifest themselves in GC skew (*Figure 6*). The ability to detect these evolutionary footprints is exquisitely sensitive to the precision with which DNA sequences can be aligned to the site of replication initiation, where the bias in Gs to Cs reverses

**Table 1.** Estimates of total number of origins by different criteria (estimates from all but first two rows rely on candidate MCM binding sites [CMBSs] from this report).

| Criterion | Estimate of number of origins |
| --- | --- |
| SGD classification | 354 (343 of which map unambiguously) |
| OriDB classification | 829 (410 confirmed, 216 likely, 203 dubious) |
| Generation of ssDNA in HU | 1600 |
| BrdU incorporation | 1400 |
| Nucleosome-free regions | 5000 |
| Transition from mostly intergenic to mostly genic | 6000 |
| Transition from low transcription genic to high transcription genic | 5000 |
| ACS motif | 6000 |
| GC skew | 6000 |

(*Figure 6—figure supplement 1*). Finally, we were able to use the power of averaging signals from multiple Mcm binding sites to extract not only direct (ssDNA) and indirect (GC skew) metrics of replication, but also to discern other characteristics of replication origins, namely their appearance in intergenic NFRs (*Figures 3 and 4*) with flanking ACS signals (*Figure 5*).

It is notable that multiple criteria converge on the conclusion that the total number of replication origins is approximately 5500 (*Table 1*): For example, in *Figure 3A* (left heat map), one can see the chemical cleavage nucleosomes signature transitioning from relatively sharp to fuzzy at rank 5000. This transition is recapitulated in the MNase-seq-based nucleosome assay (*Figure 3—figure supplement 1*) and in the MCM-ChEC signal itself (*Figure 3A*, right heat map). Furthermore, it is in this same general region, i.e., from ranks 4000 to 6000, that: (1) CMBSs transition from being mostly intergenic to mostly genic (*Figure 4A*); (2) transcription levels for genes that contain CMBSs rise sharply (*Figure 4A*); (3) position weight matrix scores for ACSs decrease sharply (*Figure 5B*); and (4) transitions in GC skew across peak midpoints disappear (*Figure 6B*). Of course, we do not conclude that all CMBS with ranks lower than 5500 function as replication origins, nor that none with ranks above 5500 do so, but only that the number of replication origins is likely to be approximately an order of magnitude higher than widely believed. Furthermore, it is possible that, as one moves to lower abundance groups of CMBSs within the most abundant 5500 sites, a smaller fraction of sites within those groups have any origin function at all. If one takes this model to the extreme, it would suggest that the continuous decline in replication activity seen in *Figure 2B* between the group comprised of ranks 1–200 and that comprised of ranks 1401–1600 reflects an ever-increasing fraction of CMBSs with zero origin activity. At the other extreme, the decline in replication activity could be interpreted within a framework in which 100% of CMBSs in each group function as replication origins, but that their replication activity declines with rank, perhaps because continuously decreasing fractions of cells in the population contain a single double hexamer. While the truth presumably lies between these two extremes, we favor a model that tilts toward the latter view, because of the abruptness of the transition that appears around rank 5000 in: (1) nucleosomal architecture (*Figure 3A and B* and *Figure 3—figure supplement 1*); (2) intergenic versus genic localization and transcription levels (*Figure 4A*); (3) EACS position weight matrix scores (*Figure 5B*); and (4) GC skew (*Figure 6B*). By these criteria, the most abundant 5000 CMBSs appear relatively homogeneous, while still showing a gradual decline in replication activity with MCM abundance within the range of detection (1–1600). Our assumption is that the qualitative homogeneity is more consistent with a quantitative, but not qualitative, change in CMBSs with declining MCM abundance among the top 5000 CMBSs. Finally, it is important to note that, in equating Mcm binding sites with potential replication origins, we are assuming that if an Mcm double hexamer is loaded onto the DNA, then it is conceivable that that complex can be activated. With approximately 6000 genes in the yeast genome, this would suggest that replication can initiate, if only rarely, in essentially every intergenic region. Cumulatively, 500 most abundant peaks of ChEC signal, which contain most of the known replication origins, comprise 85% of the abundance

of total 5500 peaks, which we propose serve as likely replication origins. This distribution of MCM abundance is consistent with that of replication initiation from two single-molecule nanopore-based studies, which found that 9% (*Hennion et al., 2020*) and 20% (*Müller et al., 2019*) of replication initiates outside the known replication origins. Our study also suggests that ~15% of replication initiation is distributed among 5000 sites, which can explain why most of these initiation events appear unique in single-molecule studies.

In addition to their ramifications about the number of sites from which replication initiates, our results have implications regarding the mechanism by which this occurs at individual sites. First, the high resolution of our approach allowed us to demonstrate that Mcm is typically loaded downstream of the ACS, something that has been demonstrated in vitro, but that researchers have thus far been unable to confirm in vivo (*Belsky et al., 2015*; *Dukaj and Rhind, 2021*). Specifically, in 112 out of 146 instances in which a peak of Mcm signal was within 100 base pairs of a known ACS, that peak was downstream of the ACS. The 34 exceptions may reflect: (1) incorrect identification of the ACS; (2) incorrect inference of the directionality of the site; or (3) sliding of the Mcm complex after it has been loaded. In interpreting these results, it is important to keep in mind that the relationship between the theoretical consensus sequence for Orc binding (i.e. the ACS) and the actual sequence to which Orc binds is so weak that approximately half of the origins reported in SGD do not have a corresponding ACS specified. Second, while we observed a general increase in GC skew across Mcm binding sites, there was a short but obvious reversal in the trend precisely at the midpoint (*Figure 6A*). This feature is consistent with the in vitro observation that Mcm single hexamers are loaded such that they must cross over each other before diverging in bidirectional replication (*Douglas et al., 2018*). Given that both the location and the size of the inflection are consistent with this hypothesis, we favor this explanation. Finally, as we have noted previously, peaks of Mcm binding typically reflect the binding of just one double hexamer, with no trend toward larger numbers of double hexamers being bound at a single origin in a single cell when moving toward higher activity origins (*Figure 3A*, Mcm-ChEC heat map). This is inconsistent with models in which higher origin activity is achieved by loading of more Mcm complexes in the same cell (*Das et al., 2015*; *Das and Rhind, 2016*; *Dukaj and Rhind, 2021*). In summary, we conclude that: (1) replication at individual origins occurs as predicted in vitro, with Mcm complexes loaded downstream of the ACS in a configuration such that they must cross over each other; and (2) part of control of origin activity that operates at the level of origin licensing is exerted by modulation of the fraction of cells in a population that have a single double hexamer bound to an origin, rather than the through variation of the number of double hexamers that are bound to individual sites in a single cell.

One potentially unexpected aspect of the change in GC skew that we observed across Mcm binding sites was the fact that the pattern was visible not just among the most abundant 1100 member cohort CMBSs, which included the most active origins, but continued to be evident, though to a lesser degree, all the way down to the fifth cohort (ranks 4401–5500), comprising the least active origins. We suggest that either or both of the following possibilities may cause even very low activity origins to leave an evolutionary footprint of replication initiation: (1) the depletion of cytosines through deamination-mediated conversion to thymines may function as a unidirectional 'ratchet', fixing changes to permanence even if they occur only rarely; and (2) the relative activity of different yeast origins may be in continual flux over evolutionary time frames, such that origins that are currently barely active were not always so.

Over the last 40 years, the field of eukaryotic DNA replication has been built largely on a foundation of discoveries made in budding yeast. Yeast's strength as a model organism came in large part because it made available short DNA sequences capable of serving as efficient replication origins. On the other hand, the sharply focused nature of its replication origins made S phase in yeast appear distinct from that in other organisms. Although by no means eliminating this distinction, our discovery that sites of replication initiation in yeast are much more widely dispersed than previously believed, with at least 1600 and possibly as many as 5500 origins, emphasizes yeast's continued relevance to understanding S phase in humans.

## Materials and methods

**Key resources table**

| Reagent type (species) or resource | Designation | Source or reference | Identifiers | Additional information |
|---|---|---|---|---|
| Strain, strain background (*Saccharomyces cerevisiae*, S288c) | 16535 | Antonio Bedalov, Fred Hutch Cancer Center | 16535 | Mat A, his3, leu2, met15, ura3, hmla:: HYG |
| Strain, strain background (*Saccharomyces cerevisiae*, S288c) | 16747 | Antonio Bedalov, Fred Hutch Cancer Center | 16747 | Mat A, his3, leu2, met15, ura3, hmla:: HYG, with MCM2 MNase 3xFlag Tag with KanMx |
| Strain, strain background (*Saccharomyces cerevisiae*, S288c) | 16753 | Antonio Bedalov, Fred Hutch Cancer Center | 16753 | Mat A, his3, leu2, met15, ura3, hmla:: HYG, with MCM6 MNase 3xFlag Tag with KanMx |
| Strain, strain background (*Saccharomyces cerevisiae*, S288c) | 16754 | Antonio Bedalov, Fred Hutch Cancer Center | 16754 | Mat A, his3, leu2, met15, ura3, hmla:: HYG, with MCM6 MNase 3xFlag Tag with KanMx |
| Strain, strain background (*Saccharomyces cerevisiae*, S288c) | 16749 | Antonio Bedalov, Fred Hutch Cancer Center | 16749 | Mat A, his3, leu2, met15, ura3, hmla:: HYG, with MCM4 MNase 3xFlag Tag with KanMx |
| Strain, strain background (*Saccharomyces cerevisiae*, S288c) | 16964 | Antonio Bedalov, Fred Hutch Cancer Center | 16964 | Mat A, his3, leu2, ura3, met15, hmla::NAT, MCM2 3x Flag Mnase tag with KanMX |

## Strain and plasmid construction and growth conditions

HU arrest in budding yeast was carried out by adding 200 mM HU to logarithmically growing cultures for 50 min.

## Sequencing

Sequencing was performed using an Illumina HiSeq 2500 in Rapid mode employing a paired-end, 50 base read length (PE50) sequencing strategy. Image analysis and base calling was performed using Illumina's Real Time Analysis v1.18 software, followed by 'demultiplexing' of indexed reads and generation of FASTQ files, using Illumina's bcl2fastq Conversion Software v1.8.4.

## Sequence alignment and quantitation

For sequence analysis, fastq files were aligned to the sacCer3 genome assembly with bwa using the -n 1 option, which causes reads that map to more than one location to be randomly assigned to one of those locations. bam files were then processed with Picard's CleanSam, SortSam, FixMateInformation, AddOrReplaceReadGroups, ValidateSamFile tools, and MergeSamFiles (version 2.21.6; http://broadinstitute.github.io/picard/). Per-base pair read depths were determined with BedTools' genomecov tool, version 2.29.1, using the -d and -split options (*Quinlan and Hall, 2010*). Quantitation of library fragments within specific size ranges was done by using the map locations of paired-end fragments to infer the insert size, and then summing per-base pair read depths across the entire inferred fragment or summing midpoints of inferred fragments, as described. This was done with the following three Perl scripts, available in the Supplemental Online Material: (1) step_1_in_custom_quantification_of_bwa_aligned_reads_121222_1.pl; (2) step_2_if_quantifying_entire_lengths_of_library_inserts_121222_1.pl; (3) step_2_if_quantifying_midpoints_of_library_inserts_121222_1.pl.

## Chromatin endogenous cleavage

ChEC-seq was carried out as previously described (*Foss et al., 2019*; *Foss et al., 2021*). Briefly, cells were centrifuged at 1500 × *g* for 2 min, and washed twice in cold Buffer A (15 mM Tris pH 7.5, 80 mM KCl, 0.1 mM EGTA) without additives. Washed cells were carefully resuspended in 570 µL Buffer A with additives (0.2 mM spermidine, 0.5 mM spermine, 1 mM PMSF, ½ cOmplete ULTRA protease inhibitors tablet, Roche, per 5 mL Buffer A) and permeabilized with 0.1% digitonin in 30°C water bath for 5 min. Permeabilized cells were cooled at room temperature for 1 min and 1/5th of cells were transferred in a tube with freshly made 2× stop buffer (400 mM NaCl, 20 mM EDTA, 4 mM EGTA)/1% SDS solution for

undigested control. Micrococcal nuclease was activated with 5.5 µL of 200 mM CaCl$_2$ at various times (30 s, 1 min, 5 min, and 10 min) and the reaction stopped with 2× stop buffer/1% SDS. Once all time points were collected, proteinase K was added to each collected time points and incubated at 55°C water bath for 30 min. DNA was extracted using phenol/chloroform and precipitated with ethanol. Micrococcal nuclease digestion was analyzed via gel electrophoresis prior to proceeding to library preparation. Library was prepared as previously described using total DNA, without any fragment size selection (*Foss et al., 2019*; *Foss et al., 2021*).

## Peak identification and quantitation

Peaks were identified by applying the following algorithm to the bam/sam alignment file that was generated by merging 12 individual bam/sam files, as described in 'Sequence alignment and quantitation': (1) quantify 51–100 base pair library inserts both across entire length and also using only the fragment midpoints, as described above; (2) smooth both data sets with a 10 base pair sliding window; (3) identify the genomic coordinate with the highest smoothed signal quantified across entire library insert; (4) choose the coordinate with the highest signal for smoothed data quantified only at midpoints that is within 60 base pairs of coordinate chosen in step 3 and assign this peak rank 1; (5) assign all coordinates in both data sets (i.e. the data set quantified along entire insert length and that quantified only by midpoint of library insert) that are within 500 base pairs of the coordinate chosen in step 4 to 0; (6) repeat the process to assign peak ranks 2, 3, etc., continuing until sharp drop-off (*Figure 1D*) indicates entire genome covered; (7) eliminate all but a single peak, which corresponds to the known rDNA origin ARS1200-1, in the highly repetitive rDNA; (8) eliminate peaks that are within 100 base pairs of Ty elements listed in SGD; and (9) renumber ranks to range consecutively from 1 (highest abundance) to 17,618 (lowest abundance). Steps 3 and 4 were motivated by a desire to choose peaks initially based on wider and presumably more robust peaks (step 3), and then to identify precise coordinates based on sharper but less robust peaks (step 4). In practice, the mean and median absolute distances that peak midpoints were shifted from step 3 to step 4 were 17.2 and 16 base pairs, respectively. Step 5 was included to avoid redundant peak assignments.

## ssDNA- and BrdU-based assessment of replication activity

Relative S/G1 ratios of ssDNA and BrdU incorporation data were taken from processed files associated with previous reports (*Feng et al., 2006*; *Yoshida et al., 2014*).

## MNase-Seq

We carried out MNase-Seq as previously described (*Foss et al., 2017*). Briefly, cells grown to log phase in rich medium, Yeast Peptone Agar with 2% glucose (YEPD), from an overnight 25 mL culture were synchronized with 3 µM alpha factor for 1.5 hr at 30°C. Arrested cells were crosslinked with 1% formaldehyde for 30 min at room temperature water bath with shaking. Formaldehyde was quenched with 125 mM glycine and cells were centrifuged at 3000 rpm for 5 min. Cells were washed twice with water and resuspended in 1.5 mL Buffer Z (1 M sorbitol, 50 mM Tris-HCl pH 7.4) with 1 mM beta-mercaptoethanol (1.1 µL of 14.3 M beta-mercaptoethanol diluted 1:10 in Buffer Z) per 25 mL culture. Cells were treated with 100 µL 20 mg/mL zymolyase at 30°C for 20–30 min depending on cell density. Spheroplasts were centrifuged at 5000 rpm for 10 min and resuspended in 5 mL NP buffer (1 M sorbitol, 50 mM NaCl, 10 mM Tris pH 7.4, 5 mM MgCl$_2$, 1 mM CaCl$_2$) supplemented with 500 µM spermidine, 1 mM beta-mercaptoethanol, and 0.075% NP-40. Nuclei were aliquoted in tubes with varying concentrations of micrococcal nuclease (Worthington), mixed via tube inversion, and incubated at room temperature for 20 min. Chromatin digested with 1.9–7.5 U micrococcal nuclease per 1/5th of spheroplasts from a 25 mL culture yielded appropriate mono-, di-, tri-nucleosome protected fragments for next-generation sequencing. Digestion was stopped with freshly made 5× stop buffer (5% SDS, 50 mM EDTA) and proteinase K was added (0.2 mg/mL final concentration) for an overnight incubation at 65°C to reverse crosslinking. DNA was extracted with phenol/chloroform and precipitated with ethanol. Micrococcal nuclease digestion was analyzed via gel electrophoresis prior to proceeding to library preparation. Sequencing libraries were prepared as described above for ChEC.

## Analysis of nucleosome chemical cleavage data

Numbers were derived from 46 to 56 base pair library fragments from bwa-aligned SRR5399542 fastq files (*Chereji et al., 2018*).

## Analysis of genic versus intergenic regions

Coordinates for gene bodies were based on the 'SGD_features.tab' file, available from https://www.yeastgenome.org.

## EACS-PWM analysis

EACS-PWM scores were based on the following position-weight matrix (from *Coster and Diffley, 2017*).

- Position A C G T
- 1 0.288 0.097 0.052 0.563
- 2 0.451 0.021 0.052 0.476
- 3 0.236 0.028 0.028 0.708
- 4 0.253 0.007 0.097 0.642
- 5 0.052 0.000 0.000 0.948
- 6 0.038 0.007 0.003 0.951
- 7 0.160 0.073 0.007 0.760
- 8 0.969 0.000 0.000 0.031
- 9 0.087 0.177 0.000 0.736
- 10 0.288 0.056 0.576 0.080
- 11 0.066 0.000 0.017 0.917
- 12 0.000 0.000 0.000 1.000
- 13 0.000 0.007 0.000 0.993
- 14 0.517 0.000 0.014 0.469
- 15 0.021 0.056 0.708 0.215
- 16 0.076 0.076 0.295 0.552
- 17 0.104 0.125 0.045 0.726

## Analysis of GC skew

GC skew was calculated according to the numbers of Gs and Cs in the Watson strand along a 51 base pair moving window, as (G-C)/(G+C).

## Acknowledgements

We thank David MacAlpine for insightful comments on the manuscript. Research reported in this publication was supported by the National Institute of General Medical Sciences of the National Institutes of Health under Award Number R01GM117446.

## Additional information

### Funding

| Funder | Grant reference number | Author |
|---|---|---|
| National Institute of General Medical Sciences | R01GM117446 | Antonio Bedalov |

The funders had no role in study design, data collection and interpretation, or the decision to submit the work for publication.

### Author contributions

Eric J Foss, Conceptualization, Formal analysis, Investigation, Visualization, Writing – original draft, Writing – review and editing; Carmina Lichauco, Tonibelle Gatbonton-Schwager, Sara J Gonske, Brandon Lofts, Uyen Lao, Investigation, Methodology; Antonio Bedalov, Conceptualization, Resources,

Formal analysis, Supervision, Funding acquisition, Investigation, Methodology, Writing – original draft, Project administration, Writing – review and editing

### Author ORCIDs

Eric J Foss http://orcid.org/0000-0002-5553-9412
Antonio Bedalov https://orcid.org/0000-0002-7373-8255

Reviewer #1 (Public Review): https://doi.org/10.7554/eLife.88087.4.sa1
Reviewer #2 (Public Review): https://doi.org/10.7554/eLife.88087.4.sa2
Author Response https://doi.org/10.7554/eLife.88087.4.sa3

## Additional files

### Supplementary files

• MDAR checklist

• Source code 1. Perl script to run as first step in quantifying paired-end reads such that one infers the entire library insert.

• Source code 2. Perl script to run as second step in quantifying paired-end reads such that one infers the entire library insert. This will provide quantitation along the entire library insert.

• Source code 3. Perl script to run as second step in quantifying paired-end reads such that one infers the entire library insert. This will provide quantitation for only the midpoint of each library insert.

### Data availability

All high-throughput sequencing data have been submitted to the NCBI Gene Expression Omnibus (GEO; https://www.ncbi.nlm.nih.gov/geo/) under accession number GSE242131.

The following dataset was generated:

| Author(s) | Year | Dataset title | Dataset URL | Database and Identifier |
| --- | --- | --- | --- | --- |
| Foss EJ, Lichauco C, Gatbonton-Schwager T, Gonske SJ | 2023 | Identification of 1600 replication origins in *S. cerevisiae* | https://www.ncbi.nlm.nih.gov/geo/query/acc.cgi?acc=GSE242131 | NCBI Gene Expression Omnibus, GSE242131 |

The following previously published dataset was used:

| Author(s) | Year | Dataset title | Dataset URL | Database and Identifier |
| --- | --- | --- | --- | --- |
| Feng W, Collingwood D, Boeck ME, Fox LA, Alvino GM, Fangman WL, Raghuraman MK, Brewer BJ | 2006 | Single stranded DNA formation during S phase in the presence of hydroxyurea in *S. cerevisiae* and *S. pombe* | https://www.ncbi.nlm.nih.gov/geo/query/acc.cgi?acc=GSE4099 | NCBI Gene Expression Omnibus, GSE4099 |

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
