## [Editor Report · eLife assessment]

This study represents a **valuable** addition to the understanding of the DNA replication origin selection process in the budding yeast. The authors provide **convincing** evidence that the number of possible origins of replication is much higher than previously appreciated, although many of the newly identified origins are likely to only direct replication initiation rarely. This work will be of interest to those studying DNA replication and investigating protein-DNA interactions across the genome.

---

## [Referee Report · Reviewer #1 (Public Review)]

The authors have previously employed micrococcal nuclease tethered to various Mcm subunits to the cut DNA to which the Mcm2-7 double hexamers (DH) bind. Using this assay, they found that Mcm2-7 DH are located on many more sites in the *S. cerevisiae* genome than previously shown. They then demonstrated that these sites have characteristics consistent with origins of DNA replication, including the presence of ARS consensus sequences, the location of very inefficient sites of initiation of DNA replication in vivo, and for the most part are free of nucleosomes. They contain a G-C skew and they locate to intergenic regions of the genome. The authors suggest, consistent with published single molecule results, that there are many more potential origins in the *S. cerevisiae* genome than previously annotated, but also conclude that many of the newly discovered Mcm2-7 DH are very infrequently used as active origins of DNA replication.

The results are convincing and are consistent with prior observations. The analysis of the origin associated features is informative.

---

## [Referee Report · Reviewer #2 (Public Review)]

By mapping the sites of the Mcm2-7 replicative helicase loading across the budding yeast genome using high-resolution chromatin endogenous cleavage or ChEC, Bedalov and colleagues find that these markers for origins of DNA replication are much more broadly distributed than previously appreciated. Interestingly, this is consistent with early reconstituted biochemical studies that showed that the ACS was not essential for helicase loading in vitro (e.g. Remus et al., 2009, PMID: 19896182). To accomplish this, they combined the results of 12 independent assays to gain exceptionally deep coverage of Mcm2-7 binding sites. By comparing these sites to previous studies mapping ssDNA generated during replication initiation, they provide evidence that at least a fraction of the 1600 most robustly Mcm2-7-bound sequences act as origins. A weakness of the paper is that the group-based (as opposed to analyzing individual Mcm2-7 binding sites) nature of the analysis prevents the authors from concluding that all of the 1,600 sites mentioned in the title act as origins. The authors also show that the location of Mcm2-7 location after loading are highly similar in the top 500 binding sites, although the mobile nature of loaded Mcm2-7 double hexamers prevents any conclusions about the location of initial loading. Interestingly, by comparing subsets of the Mcm2-7 binding sites, they find that there is a propensity of at least a subset of these sites to be nucleosome depleted, to overlap with at least a partial match to the ACS sequence (found at all of the most well-characterized budding yeast origins), and a GC-skew centered around the site of Mcm loading. Each of these characteristics is related to previously characterized *S. cerevisiae* origins of replication.

Overall, this manuscript greatly broadens the number of sites that are capable of loading Mcm2-7 in budding yeast cells and shows that a subset of these additional sites act as replication origins. Although these studies show that the sequence specificity of *S. cerevisiae* replication origins still sets it apart from metazoan origins, the ability to license and initiate replication from sites with increasing sequence divergence suggests a previously unappreciated versatility.

---

## [Author Response]

The following is the authors’ response to the previous reviews.

**Public Reviews:**

**Reviewer #1 (Public Review):**
The authors have previously employed micrococcal nuclease tethered to various Mcm subunits to the cut DNA to which the Mcm2-7 double hexamers (DH) bind. Using this assay, they found that Mcm2-7 DH are located on many more sites in the *S. cerevisiae* genome than previously shown. They then demonstrated that these sites have characteristics consistent with origins of DNA replication, including the presence of ARS consensus sequences, the location of very inefficient sites of initiation of DNA replication in vivo, and for the most part are free of nucleosomes. They contain a G-C skew and they locate to intergenic regions of the genome. The authors suggest, consistent with published single molecule results, that there are many more potential origins in the *S. cerevisiae* genome than previously annotated, but also conclude that many of the newly discovered Mcm2-7 DH are very infrequently used as active origins of DNA replication.The results are convincing and are consistent with prior observations. The analysis of the origin associated features is informative.Specific Comments:1. Page 8. The addition of an estimate of the most active origins using Southern blotting is fine for highly active origins, but how was Southern blotting used to calculate that 1-2% of cells in the eight cohort have an Mcm complex loaded.

We used a combination of Southern blotting and qPCR to measure licensing at the most active origins and then used our abundance curve to extrapolate these values to the less abundant cohorts. We expand on this point below, and we have changed the text to clarify this issue.

**Reviewer #3 (Public Review):**
By mapping the sites of the Mcm2-7 replicative helicase loading across the budding yeast genome using highresolution chromatin endogenous cleavage or ChEC, Bedalov and colleagues find that these markers for origins of DNA replication are much more broadly distributed than previously appreciated. Interestingly, this is consistent with early reconstituted biochemical studies that showed that the ACS was not essential for helicase loading in vitro (e.g. Remus et al., 2009, PMID: 19896182). To accomplish this, they combined the results of 12 independent assays to gain exceptionally deep coverage of Mcm2-7 binding sites. By comparing these sites to previous studies mapping ssDNA generated during replication initiation, they provide evidence that at least a fraction of the 1600 most robustly Mcm2-7-bound sequences act as origins. A weakness of the paper is that the group-based (as opposed to analyzing individual Mcm2-7 binding sites) nature of the analysis prevents the authors from concluding that all of the 1,600 sites mentioned in the title act as origins. The authors also show that the location of Mcm2-7 location after loading are highly similar in the top 500 binding sites, although the mobile nature of loaded Mcm2-7 double hexamers prevents any conclusions about the location of initial loading. Interestingly, by comparing subsets of the Mcm2-7 binding sites, they find that there is a propensity of at least a subset of these sites to be nucleosome depleted, to overlap with at least a partial match to the ACS sequence (found at all of the most well-characterized budding yeast origins), and a GC-skew centered around the site of Mcm loading. Each of these characteristics is related to previously characterized *S. cerevisiae* origins of replication.Overall, this manuscript greatly broadens the number of sites that are capable of loading Mcm2-7 in budding yeast cells and shows that a subset of these additional sites act as replication origins. Although these studies show that the sequence specificity of *S. cerevisiae* replication origins still sets it apart from metazoan origins, the ability to license and initiate replication from sites with increasing sequence divergence suggests a previously unappreciated versatility.Specific points:1. The authors need to come up with a consistent name for loaded Mcms at an origin. In the manuscript they variously use 'MCM'(page 3), 'Mcm complexes' (page 4), 'MCM double hexamer' (page 6), and 'double-helicase' (page 8) to describe the Mcm2-7 complexes detected in their ChEC experiments. They should pick one name (Mcm2-7 double hexamer or MCM double hexamer would be the most accurate and clear) and stick with it throughout the manuscript.

We appreciate the criticism and agree that consistency is important for clarity, thus we tried using the term "Mcm2-7 double hexamer" in every instance in which we refer to Mcm loaded at an origin. However, upon reading the resulting manuscript, we felt that these changes hurt readability more than they helped with clarity, so we left the manuscript in its original form.

2. The authors state that "It is notable that, when Mcm is present, it is present predominantly as a single doublehexamer (right panel of Figure 3A), and that this remains true across the entire range of abundance shown in Figure 3A." This statement would be improved by prefacing it with "Based on the size of the protected regions" or some other clarifying statement that lets the reader know what they should be looking for in the data in 3A.

We thank the reviewer for the helpful suggestion. We have added the underlined words to the text to clarify this point.

It is notable that, when Mcm is present, it is present predominantly as a single doublehexamer (based on the size of the protected region in the right panel of Figure 3A), and that this remains true across the enAre range of abundance shown in Figure 3A.

3. The revised statements that "We have previously used Southern blotting to demonstrate that approximately 90% of the DNA at one of the most active known origins (ARS1103) is cut by Mcm-MNase (Foss et al., 2021), and to thereby infer that 90% of cells have a double- helicase loaded at this origin. Using this as a benchmark, we estimate that ~1-2% cells have an Mcm complex loaded at the Mcm binding sites in the eighth cohort (ranks 1401- 1600)." partially clarifies how the authors came to the 1-2% number, however, the calculation is still unclear. Based on Figure 1A, there are at least three logs (1,00 fold) difference in the number of CBMSs between the best origins (which is what they state the 90% comes from) to anywhere close to the 1400-1600 rank. Seems like the number should be at best 0.1% and probably less. Either way, the authors need to explain this calculation either in the text or in the text. This sort of number tends to get thrown around later and without a clear explanation readers cannot evaluate its credibility.

We apologize for insufficiently clarifying how we arrived at our estimate of licensing. We believe that we have now remedied this, both by incorporating more measurements of licensing to improve our accuracy and by expanding the text to make our calculation unambiguous. We have added a supplemental figure showing the linear regression, based on 7 qPCR-based measurements of licensing, that we used to determine the median level of licensing of the first cohort of 200, and the altered text in the main text reads as follows:

We have previously used Southern blotting to demonstrate that approximately 90% of the DNA at one of the most active known origins (ARS1103) is cut by Mcm-MNase (Foss et al. 2021), and to thereby infer that 90% of cells have a double-helicase loaded at this origin. Combining this measurement with 6 additional measurements of licensing in cohort 1, we used linear regression (r2=0.7) to infer a median value of 69% for cohort 1. Because the median abundance in the 8th cohort is 1.5% of that in the first cohort, we estimate that CMBSs in the 8th cohort are typically licensed in 1% of cells in the population (69% x 0.015 = 1.0%).

4. The authors make the point in the introduction and discussion that recent single-molecule studies of replication origins indicate that as many as 20% of the origins identified are outside of known origins. This is very interesting but there seems to be a missed opportunity of comparing the location of these origins with the CBMSs. It would improve the manuscript to include some sort of comparison rather than using only the much older and less accurate ssDNA analysis.

Unfortunately, coverage and resolution with nanopore-based single-molecule precludes such an analysis.

5. The authors state at the end of the first paragraph on page 6 that the ChEC data is "very reproducible" which does seem to be the case but it is a little confusing for the knowledgeable reader since one would expect quite different results for an HU arrested strain versus a asynchronous or G1 arrested strain. This is hidden in the analysis in Figure S1 since 13 experiments are compared against one in each plot, however, if one x one comparisons were done there would certainly be substantial differences (or if there are not, there is a problem with the data - e.g. HU arrested cells should lack licensing at early firing origins).

It may appear counterintuiAve that one could obtain high r2 values when comparing G1 and HU-arrested samples. However, HU arrest was performed by transferring log phase cultures to 200 mM HU and harvesting cells after just 50 minutes. In this situation, most cells will be in G1 or very early S phase. Presumably, increasing times of incubation in HU would cause r2 values to decline.

6. On page 8 the authors state, "First, clear peaks of ssDNA extend down to the eighth cohort..." This seems to be stretching the data. There are clear peaks for the first five cohorts and then there is a notable change with any peak being much broader, extending over at least 10,000 bp. The authors should reconsider their statement here as it is not well supported by the data.

We have softened our language to the following: First, peaks of ssDNA signal, as judged by higher signal at the midpoints than the edges, extend down to the eighth cohort (brown line), which corresponds to CMBSs ranked 1401-1600.

7. There is one last missing reference. Wherever Eaton et al, 2010 is referenced Berbenetz, et al, 2010 (full ref below) should also be referenced as they come to very similar conclusions.

Berbenetz, N. M., Nislow, C. & Brown, G. W. Diversity of eukaryotic DNA replication origins revealed by genome-wide analysis of chromatin structure. PLoS Genet 6, (2010).

We have added this reference at all 4 instances in which we reference Eaton et al., 2010.

**Recommendations for the authors:**

**Reviewer #3 (Recommendations For The Authors):**
There are missing references in several places:

All references are included, and the references in point 3 have been split according to the reviewer's suggestion.

1. "For example, 15 of the 56 genes that contained a high abundance site have been implicated in meiosis and sporulation and are not expressed during vegetative growth (~5 out of 56 expected from random sampling), consistent with previous observations (Mori and Shirahige, 2007)." Should include Blitzblau et al., 2012 (PMC3355065) which showed that Mcm2-7 loading was impacted by differences in meiotic and mitotic transcription.2. "In contrast to the low abundance sites, the most abundant 500 sites showed a preference for convergent over divergent transcription (left of vertical dotted line in Figure 4B), in agreement with a previous report (Li et al., 2014)." This preference was first pointed out in MacAlpine and Bell, 2005 (PMID: 15868424).3. "This sequence is recognized by the Origin Recognition Complex (Orc), a 6-protein complex that loads MCM (Broach et al., 1983; Deshpande and Newlon, 1992; Eaton et al., 2010; Kearsey, 1984; Newlon and Theis, 1993; Singh and Krishnamachari, 2016; Srienc et al., 1985)." This list should include a reference to Bell and Stillman, 1992 (PMID: 1579162), which first described ORC and showed that it recognized the ACS. It would also be more helpful to the reviewer to distinguish the references that identified that ACS from those concerning ORC binding to it.